# Vehicle trajectory prediction and generation using LSTM models and GANs

**Luca Rossi**[1], **Andrea Ajmar**[2], **Marina Paolanti**[1]*, **Roberto Pierdicca**[3]

**1** Dipartimento di Ingegneria dell'Informazione (DII), Universitá Politecnica delle Marche, Ancona, Italy,
**2** Interuniversity Department of Regional and Urban Studies and Planning (DIST), Politecnico di Torino,
Torino, Italy, **3** Dipartimento di Ingegneria Civile, Edile e dell'Architettura (DICEA), Universitá Politecnica delle
Marche, Ancona, Italy

* m.paolanti@univpm.it

pone.0253868

Technology: VIT University, INDIA

**Data Availability Statement:** Data are available at:
https://www.kaggle.com/luca71/fcd-data.

**Funding:** The authors received no specific funding
for this work.

## Abstract

Vehicles' trajectory prediction is a topic with growing interest in recent years, as there are
applications in several domains ranging from autonomous driving to traffic congestion pre-
diction and urban planning. Predicting trajectories starting from Floating Car Data (FCD) is a
complex task that comes with different challenges, namely Vehicle to Infrastructure (V2I)
interaction, Vehicle to Vehicle (V2V) interaction, multimodality, and generalizability. These
challenges, especially, have not been completely explored by state-of-the-art works. In
particular, multimodality and generalizability have been neglected the most, and this work
attempts to fill this gap by proposing and defining new datasets, metrics, and methods to
help understand and predict vehicle trajectories. We propose and compare Deep Learning
models based on Long Short-Term Memory and Generative Adversarial Network architec-
tures; in particular, our GAN-3 model can be used to generate multiple predictions in multi-
modal scenarios. These approaches are evaluated with our newly proposed error metrics
N-ADE and N-FDE, which normalize some biases in the standard Average Displacement
Error (ADE) and Final Displacement Error (FDE) metrics. Experiments have been con-
ducted using newly collected datasets in four large Italian cities (Rome, Milan, Naples, and
Turin), considering different trajectory lengths to analyze error growth over a larger number
of time-steps. The results prove that, although LSTM-based models are superior in unimo-
dal scenarios, generative models perform best in those where the effects of multimodality
are higher. Space-time and geographical analysis are performed, to prove the suitability of
the proposed methodology for real cases and management services.

## 1 Introduction

Sustainable transport is one of the explicitly mentioned good practices (https://sdgs.un.org/
taxonomy/term/1198) for the achievement of the Sustainable Development Goals (SDG),
mainstreamed across several goals and targets but with a specific link to SDG 11, "Make
cities and human settlements inclusive, safe, resilient and sustainable". Sustainable transport
is considered capable to "achieve better integration of the economy while respecting the

**Competing interests:** The authors have declared that no competing interests exist.

environment, improving social equity, health, the resilience of cities, urban-rural linkages, and productivity of rural areas". Furthermore, the transport sector plays a relevant role in achieving the UNFCC Paris Agreements on climate change as close to a quarter of energy-related global greenhouse gas emissions come from transport, and that these emissions are projected to grow substantially.

Among the different initiatives towards a more sustainable transport system, the systematic update and innovation of Urban Traffic Control (UTC) systems is a common action [1, 2]. Techniques based on Connected and Automated Vehicles (CAVs) are believed to have great promises in the evolution of UTC: CAVs can communicate with other vehicles (V2V), with the traffic infrastructure (V2I), or with other entities (e.g. pedestrian, cyclist, V2X).

Communications between single vehicles and the UTC is the basis to develop and implement better signal timing plans, leading not only to increase road network efficiency but also its safety, energy economics, and pollution reduction. Real-time data collection is crucial for better traffic corridor control and management [3] and can be exploited to dynamically adjust signal timing parameters resulting in more efficient utilization of intersection capacity.

CAVs capable to acquire and transmit live single-vehicle data are more and more frequent. Acquired data includes the position, normally by means of Global Navigation Satellite Systems (GNSS) instruments and techniques, and other relevant data such as speed, heading, and engine status. Furthermore, investments in smart road initiatives, optimizing among other functionalities V2I connectivity, are significantly growing. As an example, in 2018 Italy has launched a one billion euros investment plan to develop a new smart road infrastructure to be used on 2,500 km of its 26,000 km network http://www.dreamex.it/news/16/24/Italy-is-investing-one-billion-euros-on-smart-roads). And ANAS, the Italian government-owned company deputed to the construction and maintenance of Italian motorways, in the occasion of the 2021 FIS Alpine Ski World Championships, has invested 27 million euros to install a smart road infrastructure on 80 km of the existing road network (http://www.anaspercortina2021.it/smart-road-cortina-2021). In line with this, the European Commission set out the Green Deal, highlighting the importance of developing smart and digitized systems for traffic management [4].

The availability of urban and traffic data, in particular Floating Car Data (FCD), has been increasing in recent years, as vehicles can be easily equipped with tracking devices. This data can be exploited for different applications, mainly autonomous driving, and traffic congestion. Furthermore, there is an increasing need for intelligent systems able to predict short-term and long-term trajectories of vehicles in a smart road context. Trajectory prediction can be exploited to minimize vehicle travel time and meanwhile avoid generating congestion, developing methods for depicting the future utilization of the road network (e.g., vehicle density, driving time, and probability of accident) based not on the single vehicle's perspective but on all projected trajectories [5]. The long-term trajectory prediction of surrounding vehicles is essential for autonomous vehicles: for example, a vehicle equipped with trajectory prediction can not only avoid an accident but also generate evenly distributed control input sequences, such as a jerk-minimizing acceleration by reacting in advance [6]. Moreover, such models can be exploited for anomaly detection in connected automated vehicles.

With the advancement of deep learning techniques and computing power, combined with the development of new models, namely Long Short-Term Memory (LSTM) [7] for prediction and the more recent Generative Adversarial Network (GAN) [8] for generation, systems are capable of processing large amounts of data and different fields of research, including trajectory prediction, are witnessing unprecedented growth [9]. Being able to make real-time short-term predictions of vehicle trajectories, as well as long-term ones, is essential for these applications, but it comes with some challenges due to the multimodal nature of the problem: several

predictions can be equally plausible in a given scenario, there is no single correct solution for the deep model to find [10].

The focus of this paper is to address the problem of multimodality in FCD trajectory prediction. The main motivation behind this research is the lack of existing methods for approaching trajectory prediction in a multimodal fashion, which limits the scope of practical applications. To address this problem we propose a generative method that can produce multiple plausible predictions. The evaluation of unimodal methods is also distorted by the biases in the standard metrics, so we introduce new ones that normalized those biases.

We train and evaluate predictive and generative models on 4 different datasets, pointing out the advantages and disadvantages of each method. FCD used for these analyses were acquired and pre-processed by the company VEM Solutions S.p.A. during an entire week (from 2018–10-05T22:00:00 to 2018–10-11T21:59:59 CET) in the four most populous Italian cities (Milan, Naples, Rome, and Turin). VEM on-board units (OBU) are mounted on fleets and private cars with a varying penetration rate, estimated between 2 and 8%.

While predictive methods aim to minimize error with respect to ground truth, generative ones aim to produce samples that are as realistic as possible. There are two important consequences. The first is that while predictive models have only one solution, generative ones can have several. The second is that predictive models tend to predict an "average behavior", while generative ones can reproduce real ones. This distinction is fundamental because of the *multimodal* nature of the problem. While it may be impossible to predict a trajectory in highly uncertain scenarios (i.e., the correct solution cannot be found), it is still possible to generate (i.e., enumerate) many plausible solutions that do not necessarily minimize prediction error. We associate the prediction problem with unimodal scenarios and the generation problem with multimodal scenarios.

In the evaluation of our experiments, we use standard metrics for trajectory prediction, i.e. the Average Displacement Error (ADE) and the Final Displacement Error (FDE), as well as new normalized metrics that reduce the bias is unaccounted for by the standard ones. Those new metrics, i.e. the Normalized Average Displacement Error (N-ADE) and the Normalized Final Displacement Error (N-FDE), are one of the main contributions of this work, alongside the datasets and models. When evaluating our multimodal GAN-3 model, we take into account its ability to correctly reproduce a behavior rather than identify the exact one among many alternatives, which is not possible in truly multimodal scenarios.

The advantage of the proposed GAN-3 model over those used by other works is its ability to represent multiple distinct behaviors instead of producing a single average prediction, this makes the predictions more realistic and unaffected by the problem of multimodality. Similarly, the advantage of the proposed metrics is that they are unaffected by biases in the standard ones that depend on properties on the trajectories and not on the predictions.

## 1.1 Challenges

Predicting vehicle trajectories is an arduous problem that requires addressing several challenges. The most relevant ones can be grouped into the following categories:

- **Vehicle to Infrastructure (V2I) interaction.** Vehicle trajectories depend on the environment, which can be the physical space (i.e. the infrastructure, or road network) or the temporal window in which they are collected.

- **Vehicle to Vehicle (V2V) interaction.** Vehicle trajectories mutually influence each other. There are two ways in which trajectories influence each other: by physically restricting the

possible trajectories of other vehicles, or by influencing their behavior as they must follow traffic rules.

- **Multimodality.** Trajectory prediction is highly uncertain because there are multiple possible behaviors and multiple correct solutions. Uncertainty has an aleatoric component, which cannot be reduced because it depends on the randomness of the system, and an epistemic part, which can be reduced by adding new information.

- **Generalizability.** A method should be evaluated on its ability to predict the entire distribution of possible vehicle trajectories, collected in a realistic environment and not in a controlled one.

Exploring all these challenges would require a lot of additional research. This paper focuses on the multimodality aspect, as we will show how predictions are influenced by high uncertainty. We will also partially tackle the problem of generalizability by proposing new realistic FCD datasets collected in four different cities, and showing how different datasets yield different results.

## 1.2 Contributions

The main contributions of this paper are the following:

- **New datasets.** We propose four new FCD datasets with a large number of vehicle trajectories. Those datasets contain a lot of information about vehicles and road networks, but for our purposes, we create a preprocessed version that only takes into account vehicle coordinates, where we split long trajectories into short ones, more practical for real-time predictions.

- **New metrics.** We propose a normalized version of the standard ADE/FDE metrics to take into account different biases in the evaluation of the models in different scenarios.

- **A generative model for multimodal scenarios.** We propose a GAN model able to generate multiple predictions and overcome some of the challenges of purely predictive models (e.g. LSTM).

- **The application of innovative data analysis and visualization techniques.** We tested space-time pattern mining tools as support methods to better understand spatiotemporal trends and to generate analysis outputs suitable to support UTC actions.

The paper is organized as follows. Section 2 provides a description of the approaches adopted when facing vehicle trajectories. Section 3 thoroughly describes our contributions, i.e. datasets, metrics, and the GAN-3 model. In Section 4, we evaluate and compare predictive and generative models on our proposed datasets with the standard metrics and our newly proposed ones. Finally, in Section 5, conclusions and discussion about future directions for this field of research are provided.

## 2 Related works

The use of FCD for traffic management and control, along with its benefits and limitations, has been widely discussed, generally mentioning the insufficient penetration rate as the main limiting factor [11]. Houbraken et al. [12] make a review of different studies related to the necessary FCD penetration rates and find that this value is dependent on the intended application and data sample frequency. Altintasi et al. in [3] find that penetration rates around 3% are sufficient to identify some critical urban traffic states.

Ajmar et al. in [13] compare floating car data (FCD) with low penetration rates with data acquired by fixed sensors. Currently, most models used by UTC are based on traffic waves and on data acquired by fixed sensors that have several issues: they record data at a specific location and, due to their installation and maintenance costs, they are not sufficiently distributed, with a reduced or nonexistent coverage especially on minor roads. This leads to the necessity of building dedicated models to estimate the traffic states at other locations and therefore of dealing with related uncertainties.

Guo et al. in [1] provide a systematic review of CAV-based UTC studies. They state that the availability of high-resolution trajectory data of individual vehicles could increase the understanding of traffic flow states, which are critical to traffic control, allowing to estimate values not only related to flow volumes but also travel time, queue length, and shockwave boundary that cannot be calculated based on fixed sensors data only.

Many works use FCD with the aim to detect and analyze traffic in an urban environment. Chen et al. in their recent work [14] use the speed performance index (SPI) to evaluate the traffic congestion based on the speed of the traffic flow and the speed limits of the road. They propose a categorization criterion to make consistent the classification results of SPI with the five levels standard of traffic congestion classification. The investigation has the aim to detect the paths of traffic congestion, finding both spatial and temporal correlations.

Similarly, Altintasi et al. in [3] use the FCD with the information on speed to detect the traffic status. They process the raw FCD to obtain 4 Levels of Service (LOS) that identify the traffic state. Each LOS corresponds to a different range of speed and traffic situation. With the experimental phase in a real urban environment, by observing the pattern variations, they are able to detect how traffic changes through time in a given location.

Sunderrajan et al. in [15] also evaluate the traffic flow departing from FCD. The authors do not only consider traffic speed, but also other aggregate macroscopic quantities such as density and flow. This was the first attempt to estimate both traffic flow and density from FCD. Moreover, an interesting novelty is the introduction of a method to determine the minimum rate of probe vehicles needed to reconstruct the traffic state in different conditions.

However, the papers presented in this section so far do not make traffic predictions on FCD but detect and recognize the situation of traffic congestion. A pioneering work in this direction is the one proposed by De Fabritiis et al. in [16]. This work exploits the real-time FCD based on GPS positions to create a system for traffic estimation and prediction. They propose a system that extracts the speed information, making an estimation of traffic conditions. Moreover, using two different approaches based on pattern matching and artificial neural networks, they are able to predict the short-term speed traffic conditions, with a maximum prediction time of 30 minutes.

More recent is the work of Kong et al. [17], which exploits the trajectories data to make predictions of traffic congestion, indicating in real-time an alternative path. The parameters used to represent the traffic flow are the traffic volume and traffic speed. The idea is to use a fuzzy comprehensive evaluation method to determine the traffic flow and predict the traffic congestion, where the weights of the indexes are assigned according to the traffic flow. These weights are determined by the Particle Swarm Optimization (PSO) algorithm, and then a congestion state fuzzy division module is applied to convert the predicted flow parameters to citizens' cognitive congestion state. The metrics used to evaluate the performance of the proposed approach are accuracy, instantaneity, and stability.

A method based on PSO is also proposed by Luo et al. in [18]. The authors present a hybrid architecture for short-term traffic flow prediction by combining PSO and a genetic algorithm. This hybrid approach is applied to the Least Square Support Vector Machine (LSSVM) to select the appropriate parameters to predict short-term traffic flow.

More recently, the most used approaches for traffic flow prediction are based on deep learning algorithms [19]. The most used deep learning method for traffic prediction is the Long Short-Time Memory (LSTM), as in [20, 21]. Vazques et al. in [22] compared four deep learning models (LSTM, GRU, SRCN, HGC-LSTM) for urban traffic prediction. The dataset they used consists contains the average speeds in different periods of the tracked vehicles. Given the state of the network in a time period, the models have to predict the state of the network in the following one. The metrics used for the comparison are the Mean Average Error (MAE) and the Root Mean Square Error (RMSE). Other works in vehicle trajectory prediction, like the DESIRE framework proposed by Lee et al. in [23], experiment on public datasets like KITTI [24] and Stanford Drone Dataset [25], that provide visual information instead of FCD.

In [26], the authors introduce and compare two different machine learning methods that use floating car trajectory data to predict the occurrences of crashes on urban expressways: a binary logistic regression model and a Support Vector Machine (SVM) model. According to the authors, the latter performs greatly for predicting crashes on urban expressways. Implementations of such models for monitoring real-time conditions seem promising in helping to detect and prevent potential crashes from happening.

Finally, some works from the domain of human trajectory prediction propose interesting methods that could be extended to the domain of vehicle trajectory prediction. For example, Alahi et al. in [27] propose the Social LSTM, a network that predicts human trajectories taking into account interactions with other human trajectories. This is the equivalent of V2V interaction for vehicles. Gupta et al. in [28] extend this work and propose the Social GAN, that generates multiple predictions in a multimodal context, similarly to our work.

## 3 Methodology

Among the four challenges outlined in the introduction, we focus on *multimodality* and partially on *generalizability*. In this Section, we provide a formal definition of the trajectory prediction problem (Section 3.1), an analysis of the proposed datasets (Section 3.2), a description of the proposed normalized metrics (Section 3.3), and the implementation details of our GAN-3 model (Section 3.4).

### 3.1 Problem definition

A *trajectory* is the route taken by a vehicle over a set period of time. Unless otherwise stated, we define it as a vector of numerical values that alternatingly correspond to the $X_i$ and $Y_i$ coordinates at increasing timestamps $i$ separated by a fixed time-step. A trajectory of $n$ points is a sequence of $2n$ values $X_1, Y_1, X_2, Y_2, \ldots, X_n, Y_n$, or an element of the set $T$ as defined in (1).

$$T = \{X_i, Y_i : i \in [1 \ldots n]\} \tag{1}$$

The trajectory point index is mapped to a timestamp by the function *time*(*x*), which respects the condition in Eq (2), with *K* being a fixed time-step.

$$time(i) - time(j) = K \cdot (i - j) \tag{2}$$

The trajectory *prediction* problem is defined as the task of estimating values of future time-steps according to the observed values of past time-steps by minimizing a loss function based on the distance from the ground truth. We define the trajectory prediction problem as the following sequence generation task: given an *observed trajectory* of $m < n$ points as a sequence $X_i, Y_i$ for each $i \in [1 \ldots m]$, predict the values of $X_i, Y_i$ for each $i \in [m + 1 \ldots n]$ by minimizing some error metrics with the actual trajectory $t \in T$.

The *generation* problem is defined as the task of synthesizing samples that can realistically fit into the original data distribution. The trajectory generation problem is similar to the prediction one, but instead of minimizing some error metrics, the goal is to find some $t_i \in T$ that realistically describe a vehicle trajectory.

It is important to clarify that the problem of trajectory prediction is different whether we consider short trajectories (a few seconds long) or long ones (minutes or hours long). These are two different problems with different challenges and applications.

Since we address the problem of multimodality, the trajectories we analyze are a few minutes long: long enough so there are different possible routes in order to make a multimodal analysis, short enough so that this analysis can be applied to contexts such as self-driving cars and traffic congestion prediction. The longer a trajectory is, the more difficult it is to predict, and it requires a different kind of analysis.

In the rest of this paper, the terms "short trajectory" and "long trajectory" will have a different meaning and refer to the precise length of the trajectories we analyze: 12 minutes for short trajectories, 20 minutes for long ones. In both cases, the observed part of the trajectory has a fixed length of 8 minutes. We expect the prediction error to be larger for long trajectories.

## 3.2 Data processing

For our analysis, we introduce new datasets obtained by preprocessing the GNSS data of vehicle trajectories from 4 different cities, already described in [13]: Rome, Turin, Milan, and Naples.

These datasets contain a vast amount of data, both in form of data points and features. They provide useful information about the type of vehicle and the road graph, but for our purposes we discard most of it and only keep coordinates and timestamps. We apply some preprocessing steps to split macro-trajectories into shorter ones with a fixed number of data points and reduce noise.

Table 1 reports the number of data points for each dataset, before and after preprocessing, as well as the number of resulting trajectories. There are four preprocessing steps:

1. Resampling: trajectories (lists of coordinates associated with the same vehicle) are resampled with a 60000 ms period, as the original sampling is irregular (with a mean of 60000 ms). This makes all the trajectories uniform time-wise and partially reduces noise.

2. Splitting: long trajectories are split into shorter ones with fixed length.

3. Normalization: coordinate values are normalized within the range [0, 1]. Predicted values are later denormalized to calculate the error metrics.

4. Filtering: idle and noisy trajectories, that don't provide useful information, are removed.

In the filtering step, *idle* and *noisy* trajectories are removed. We consider a trajectory *noisy* if it contains at least two consecutive coordinates with a difference greater than 0.1 on values normalized in the range [0, 1]. We consider a trajectory *idle* if its variance is lower than

**Table 1. Datasets and number of data points.**

| Dataset | Original data points | Preprocessed data points | Trajectories |
|---|---|---|---|
| Rome | 12,979,442 | 5,159,880 | 257,994 |
| Turin | 4,000,656 | 1,156,920 | 57,846 |
| Milan | 3,361,692 | 977,740 | 48,887 |
| Naples | 8,150,826 | 2,679,400 | 133,970 |

$\varepsilon$ = 0.00001 on coordinates normalized in the range [0, 1]. An idle trajectory must be completely idle. For example, trajectories of vehicles that start moving after having been idle for just a few time-steps are not considered idle. Removing idle trajectories is necessary because they decrease the value of error metrics without reflecting the ability of a model to predict real, non-idle trajectories.

Following these steps, we obtain two different preprocessed datasets: one with short trajectories and another with long ones. The former has 12-point trajectories (8 observed points and 4 predicted points) while the latter has 20-point trajectories (8 observed points and 12 predicted points).

## 3.3 Evaluation metrics

The standard metrics used in the field of trajectory prediction are the Average Displacement Error (ADE) and the Final Displacement Error (FDE) proposed by Pellegrini et al. [29]. ADE measures the RMSE between each pair of predicted and true trajectory points, while FDE only considers the difference between the predicted and true final points.

Although these metrics are straightforward to implement and interpret, they have some limitations when evaluating methods used in real-world scenarios. In controlled scenarios, trajectories tend to have similar properties (e.g., same length), but in real-world scenarios, these properties can greatly affect the evaluation. For example, the value of ADE tends to be higher for longer trajectories (with many data points) because farther time-steps are more unpredictable and thus the average error tends to be larger. This problem can't be solved by normalizing only once (i.e., dividing the total error by the number of time-steps), as the error doesn't grow linearly.

For example, if trajectory $T_1$ is longer than trajectory $T_2$, the average error of $T_1$ will be higher than the average error of $T_2$ because the higher uncertainty of the farthest time-steps increases the average error.

The use of ADE/FDE would generate a bias against longer trajectories and favor models that learn short trajectories well. These limitations justify the use of the new metrics defined in this work; however, ADE/FDE are still valuable and intuitive, so we use them jointly with our proposed normalized metrics N-ADE/N-FDE. In fact, N-ADE/N-FDE are not a replacement for ADE/FDE. The advantage of the normalized metrics is that they are mostly independent of trajectory properties, the disadvantage is that they can only be used in an aggregate manner since they don't provide useful information if evaluated on a single trajectory.

Many alternative ways to measure the similarity between two trajectories have been proposed over the years, and several interesting approaches, like those based on Edit Distance or Dynamic Time Warping, have been reviewed by Magdy et al. [30]. While most of these approaches address the limitations of ADE/FDE, they are more computationally expensive and lose physical meaning (i.e., they don't have a physical unit of measurement unit, such as meters). A good alternative to ADE/FDE should not only take into account the biases previously mentioned, but it should also be computationally efficient while maintaining physical meaning.

N-ADE/N-FDE are defined by normalizing ADE/FDE, to the square root of these properties multiplied by constants. The square root attenuates the effect of this normalization; it should not be completely independent of the trajectories, and the constants are only used to maintain an intuitive physical meaning.

We define the aggregates Normalized Average Displacement Error (N-ADE) and Normalized Final Displacement Error (N-FDE) as in Formulae (3) and (4).

$$NADE_P = M_t[ADE(t) \cdot NF_P(t)] \tag{3}$$

$$NFDE_P = M_t[FDE(t) \cdot NF_P(t)] \tag{4}$$

$$NF_P(t) = \prod_{p \in P} \sqrt{\frac{K_p}{p(t)}} \tag{5}$$

The normalized metrics are calculated using the parameter $P$, which is the set of chosen properties to normalize. ADE/FDE are multiplied by the Normalization Factor (NF) defined in Formula (5). *ADE(t)*, *FDE(t)*, and *NF(t)* are functions of the trajectory $t$. The NF is calculated as the inverse of the product of the chosen properties $p(t)$, multiplied by the constants $K_p$.

For our purposes, we choose the length (in meters) and the variance of the trajectory as properties to normalize, but future works could use different properties. This choice comes from the assumption that long trajectories are more difficult to predict and they lead to higher errors, as well as high-variance (noisy) ones. We redefine our *NF* as in Formula (6).

$$NF_{\{l,v\}}(t) = \sqrt{\frac{K_l}{l(t)} \cdot \frac{K_v}{v(t)}} \tag{6}$$

As for the constants, we chose the median values of the datasets, truncating the length to the integer and the non-linearity to the first decimal, as in Formulae (7) and (8) (it's worth remembering that these constants aren't necessary and are arbitrarily chosen, we only use them to make the metrics more intuitive).

$$K_l = 0.04 \tag{7}$$

$$K_v = 0.003 \tag{8}$$

The aggregate N-ADE and N-FDE of an entire dataset are calculated as the median value, which is a better choice than the mean. This is because it is more robust to very low and very high values, which are common with N-ADE/N-FDE defined using the chosen properties. The error can approach infinity if these values are too low, and the average error of a whole dataset can be huge even if there is only one faulty trajectory. This is not an issue when using the median aggregator $M_t$.

## 3.4 Implementation details

We approach the problem of vehicle trajectory prediction with two different methods that we describe here and compare in Section 4.

The first method is purely predictive and is based on an LSTM architecture. This method tries to predict the future positions of the vehicles by minimizing the Mean Squared Error loss function.

The second method has a generative nature and is based on a GAN architecture. This method tries to generate realistic trajectories that can be used to make multiple predictions. This is necessary for multimodal scenarios, where multiple correct solutions are possible. With the GAN-3 model, we generate 3 different trajectories, each representing a different behavior, then pick the correct behavior and measure the error w.r.t. that behavior.

Each of the two methods is more suitable for particular scenarios (namely, LSTM for unimodal scenarios and GAN-3 for multimodal ones). We then compare the two approaches in Section 4, where we also compare GAN-3 with its unimodal counterpart, GAN-1, which is expected to perform worse than the purely predictive LSTM.

**3.4.1 Predictive model architecture.** The chosen predictive architecture is based on the Long Short-Term Memory (LSTM) network. This model is a Recurrent Neural Network (RNN) and is best suited for problems that require learning patterns in a time series (a sequence of values corresponding to increasing timestamps with a fixed period or time-step); for this reason, it is also the standard choice in other works in this field. An LSTM network contains one or more LSTM layers that take as input a time series that describes the past and produce as output another time series that describes its future prediction. The input of those layers needs to be reshaped in such a way that one dimension represents the timesteps and another dimension represents the value of the data at each timestep. Our LSTM is stateless, which means that no memory is retained when training on different trajectories, as they do not belong to the same time series. Optimal results are achieved quickly with minimal tuning considering that the training data has a very simple representation (a sequence of 16 numbers).

The architecture of the LSTM model is very simple: it consists of two stateless LSTM layers with 32 and 16 neurons respectively, followed by a Dense layer with 16 neurons and a final Dense layer with a number of neurons corresponding to the number of timesteps to predict (that can be 4 or 12) multiplied by the number of coordinates (2). The first layer receives an input of 16 values as the 8 $(x, y)$ ordered coordinates in the observed sequence. A simple visual representation is shown in Fig 1: the observed trajectory is given as input to the Input layer, then it's passed to the Reshape layer that organizes the input in a way that groups and separates the timesteps by the $x$ and $y$ coordinates. The resulting data is fed to two consecutive LSTM layers. Finally, we have a fully connected Dense layer followed by the output, i.e. the predicted trajectory as another Dense layer.

We train our model with a batch size of 256. We use the Rectified Linear Units (ReLU) activation function. The loss function to minimize is the Mean Squared Error.

We use the Adam optimizer with an initial learning rate of 0.001. Regularization is achieved with Batch Normalization and Dropout layers set with a probability of 0.2. We do not limit the number of epochs since the models converge fast, instead, we set a *patience* of 200, which means that the training stops if no improvements have been made over the last 200 epochs.

Each of the 4 datasets (Rome, Turin, Milan, and Naples) is shuffled and split into a training set (56%), a validation set (14%), and a test set (30%). Partial experiments with cross-validation did not result in significant improvements, so we decided to use a fixed validation set to maximize training speed.

**3.4.2 Generative model architecture.** We use a Generative Adversarial Network (GAN) to generate plausible trajectories and thus produce multiple predictions in a multimodal environment. The GAN model, introduced by Goodfellow et al. in [8], consists of two networks, a Generator and a Discriminator. The Discriminator learns to distinguish between real (i.e., fitting into the original data distribution) and fake data, while the Generator aims to fool the Discriminator by producing realistic data. If the model is trained well, the Generator will be able to produce new data samples that fit into the training data distribution; in our case, the predicted trajectories.

We distinguish among two generative methods, GAN-1 and GAN-3. For simplicity, we will refer to GAN-1 and GAN-3 as two different models, while they are actually the same model and are trained in the same way. The only difference is in the testing phase.

Both the Generator and the Discriminator receive the observed trajectory as input. The Generator also takes a latent vector as input and produces the output trajectory. The Discriminator also receives the generated trajectory as input and decides whether the entire trajectory is real or fake. A simple visual representation of the GAN model is shown in Fig 2: two Input layers take the latent vector and the observed trajectory that are concatenated and fed to the

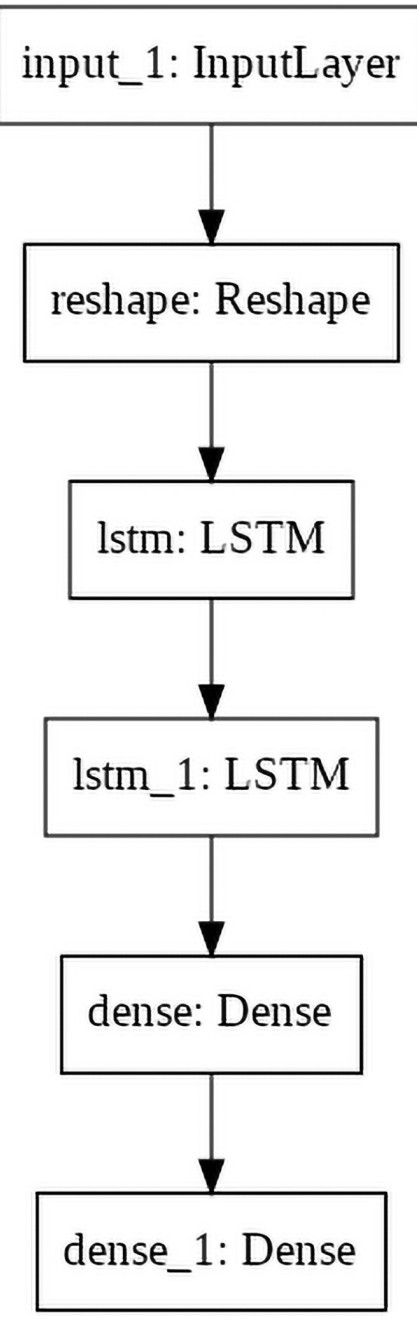

**Fig 1. Visual representation of the LSTM model graph.**

Generator layer. The Generator layer produces an output that is concatenated with the observed trajectory input to produce the input for the Discriminator layer. The Generator and Discriminator layers are exploded respectively in Fig 3a and 3b.

We use LeakyReLU instead of ReLU as the activation function of both Generator and Discriminator layers. LeakyReLU allows a small positive gradient when the neuron is not active, it helps to mitigate the vanishing gradient problem, which is particularly common in GAN models. In general, high regularization is needed when designing a GAN Discriminator to avoid

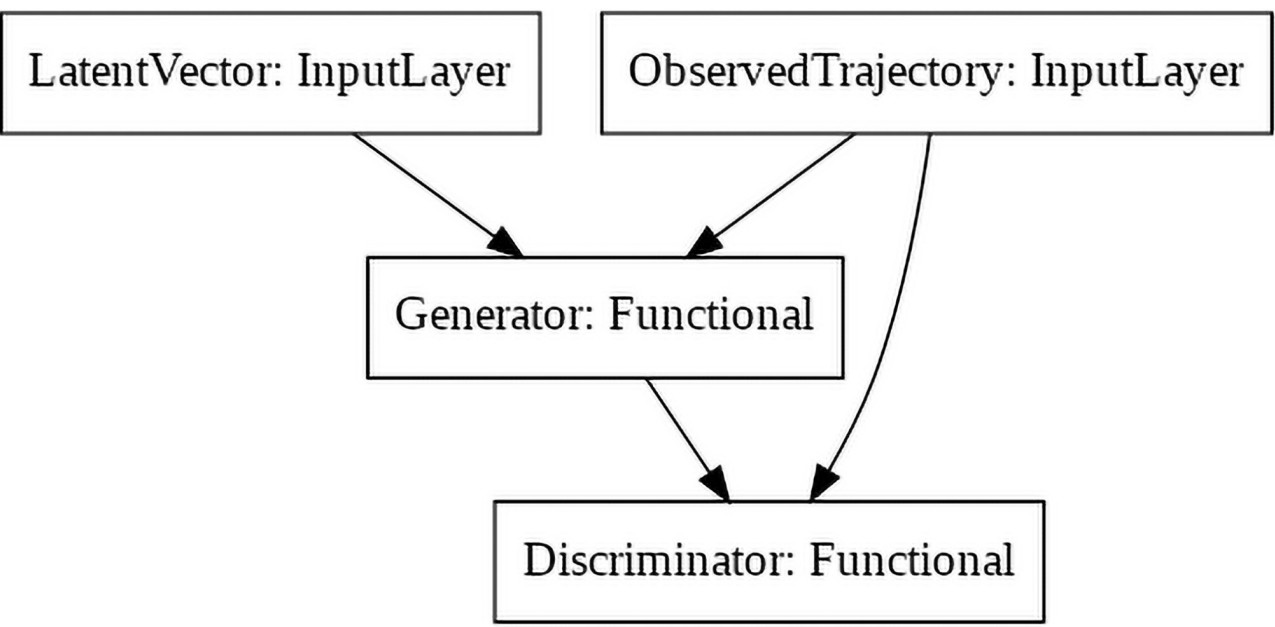

**Fig 2. Visual representation of the GAN model graph for prediction.**

mode collapse. Both the Generator and the Discriminator are heavily regularized with Batch Normalization and Dropout layers set with a probability of 0.5.

The Discriminator has to learn to distinguish real trajectories from fake ones. By contrast, the Generator has to learn to produce plausible trajectories. A simple visual representation of both models is shown in Fig 3.

To understand the architecture of the Discriminator, we can divide it into three sub-networks. The first sub-network takes as input only the trajectory produced by the Generator, and it learns some features without taking the observed trajectory into account. This sub-network learns if the Generator can produce meaningful trajectories, regardless of whether they are meaningful predictions for the observed ones. It contains an LSTM layer with 64 neurons (32 for 4-point predictions) followed by a Dense layer with 32 neurons (16 for 4-point predictions).

The second sub-network learns to model the entire trajectory. It learns if the trajectory produced by the Generator is a good prediction for the input trajectory. Its input is the concatenation between the observed trajectory and the trajectory produced by the Generator. It contains an LSTM layer with 64 neurons followed by a Dense layer with 32 neurons.

The third sub-network takes as input the concatenated outputs of the two other sub-networks and produces the final output (real or fake). It contains 3 Dense layers with 32, 16, and 1 neuron each. The activation function of the final Dense layer is the sigmoid.

Similarly, to understand the architecture of the Generator, we can divide it into two sub-networks. The first sub-network takes as input the observed trajectory and learns some basic features. This sub-network doesn't consider the latent input used for generating new samples, meaning that the output of this sub-network will always be the same for a given observed trajectory. It contains an LSTM layer with 32 neurons, followed by another LSTM layer with 16 neurons, followed by a Dense layer with 16 neurons.

The second sub-network takes as input the output of the previous one, concatenated with the latent input. The latent input allows for a mapping between the latent space and the output

(a)

(b)

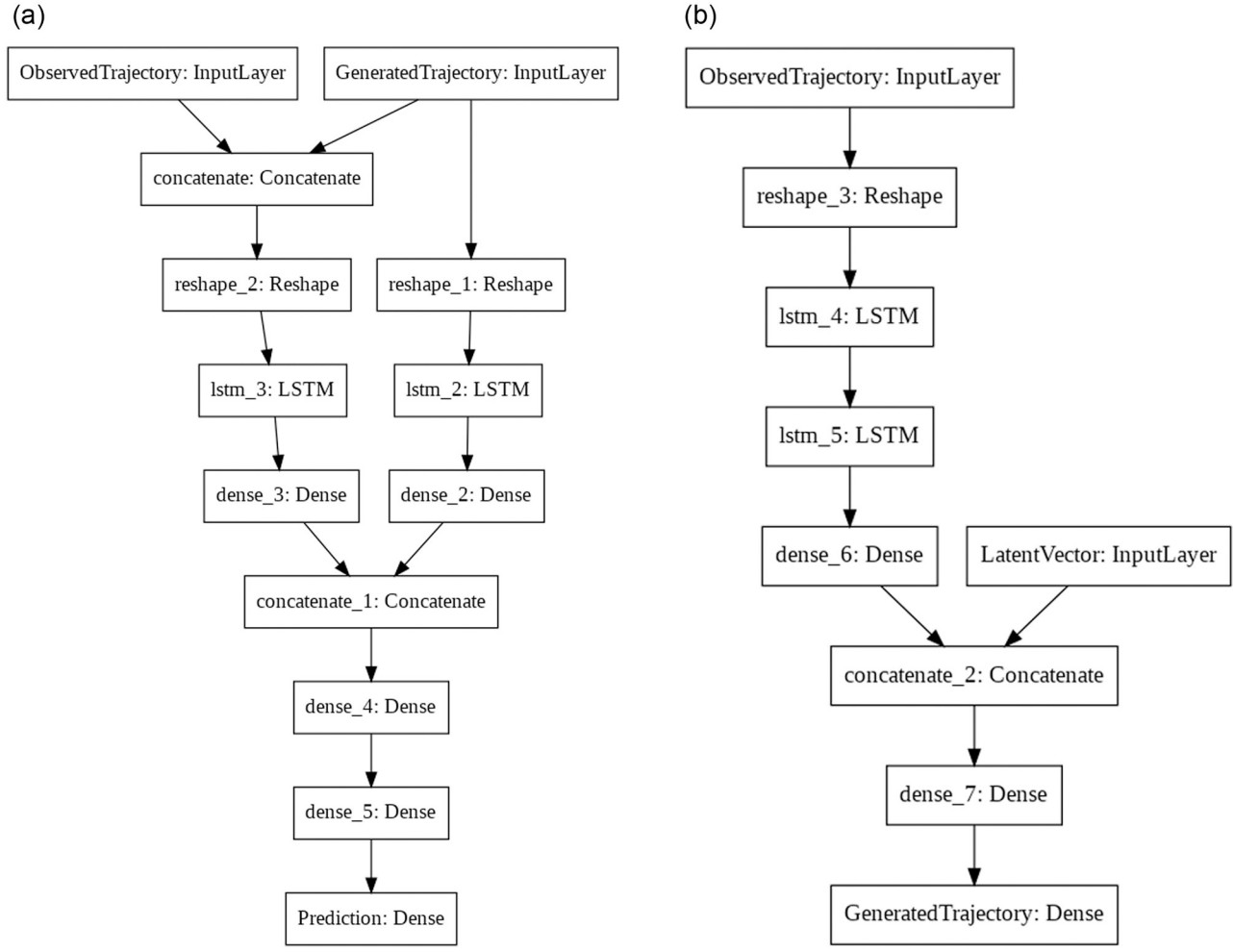

**Fig 3. Visual representation of the Generator and Discriminator models of the GAN.** Regularization (Dropout and Batch Normalization) and activation (LeakyReLU) layers are not shown for simplicity.

space, meaning that multiple predictions can be made. The latent input is a vector with 2 values, as the low dimensionality helps to avoid that the latent input overshadows the output of the first sub-network, and mitigates the risk of mode collapse. The sub-network contains a Dense layer with 16 neurons, followed by a final Dense layer with as many neurons as the number of timesteps to predict (4 or 12) multiplied by the number of coordinates (2). The activation function of the final Dense layer is the hyperbolic tangent.

The Generator produces the non-observed part of the trajectory, which is concatenated to the observed part and passed as input to the Discriminator.

When training the GAN, each epoch consists of a Generator step and a Discriminator step. When training the Generator, we feed the input to the entire GAN model while setting the Discriminator weights as not trainable. Both training steps produce the same type of output, i.e. the predicted "real" or "fake" class, but the loss function is the opposite, as the Generator aims to fool the Discriminator into classifying all its samples as "real".

The difference between using an LSTM and a GAN for prediction lies in the objective function. The LSTM learns an average behavior because it minimizes the average error between all the predictions. The GAN learns to produce realistic samples corresponding to specific

behaviors. This is why using the GAN-1 model (that only makes one prediction) in a multi-modal scenario is counterproductive; even if the model is good at generating plausible trajectories, it can produce a different behavior from the real one, resulting in high errors.

**3.4.3 Multimodal generation.**   One example of multimodal generation in trajectory prediction is the SGAN-20 model by Gupta et al. in [28] in the domain of human trajectory prediction. They propose the SGAN-20 model, which uses the Social GAN to produce 20 different samples and selects the best one by comparing the errors with the ground truth. One major problem with this approach is that it can't be compared with unimodal predictions, because part of the reason why SGAN-20 outperforms the unimodal SGAN is that it simply makes more attempts.

This problem can't be completely avoided since a better approach to address multimodality does not exist yet, but our proposed GAN-3 partially decreases the impact of chance on error decreases.

The problem with methods like SGAN-20 is that we cannot be sure that the 20 samples describe 20 different behaviors. SGAN-20 assumes that the behaviors are all different and exactly one of them is correct. This is seldom true and the results are unreliable.

GAN-3 is similar to SGAN-20 and it too picks the best result w.r.t. the ground truth, but we implement two major differences. First, we reduce the number of samples. Assuming that each sample should represent a different behavior, a number of 20 possible behaviors is excessive for trajectory prediction. Furthermore, it depends on what is considered to be a different behavior. For example, potential behaviors could include turning left, turning right, turning left and stopping, or turning left and accelerating.

The second difference is that we compare the generated samples to determine whether they represent different behaviors. The problem is that there are different levels of abstraction when considering different behaviors, so we set a heuristic to quantify the level of abstraction we want to use in our evaluation.

We approximate the *distance* between two behaviors and use it to define a threshold $T \in (0,1)$ and choose the level of abstraction that differentiates two behaviors. With a high $T$ there are just a few possible behaviors (perhaps with a simpler semantic representation), while with a low $T$ there are many possible ones (with a more complex semantic representation). If $T$ is too high, the problem becomes unimodal, so using an LSTM would be more convenient. If $T$ is too low, every generated trajectory would be a different behavior, and it would be the same as SGAN-20.

In general, we define our GAN-K model as a model with the architecture previously defined, that when evaluated produces $K$ samples. Samples are produced sequentially. When the sample $i$ is generated, it is compared with the previous $i − 1$ samples by checking the *distance* from those behaviors. If the distance is too low for at least one behavior, the sample is discarded and generated again. After a given number of attempts, that we set to 100, we stop the generation and only keep the first $i − 1$ samples.

The distance between two behaviors is calculated as the ratio between the lowest of the two errors and the highest. If this value is higher than $1 − T$, the new sample is discarded. The errors we compare are the values of the N-ADE metric. In mathematical terms, the multimodal GAN generates $K$ samples that satisfy the condition in Formula (9) given a threshold $T$ and the N-ADE error $E_i$ for the sample $i$. Our heuristic to distinguish between different behavior is to check if the ratio between the two errors is high. Intuitively, samples corresponding to the same behavior should have similar errors, although the opposite is not necessarily true. In our case, with $K = 3$, we choose threshold $T = 50\%$.

$$\frac{E_i}{E_j} <= 1 - T \qquad \forall i,j \in \{1, ..., K\}, i \neq j \tag{9}$$

To clarify, GAN-3 cannot be used for unimodal prediction, as in real-world scenarios we don't have access to the ground truth. We use this method to analyze multimodality in a given environment. Moreover, GAN-3 is a stochastic method that can potentially generate different sets of samples that satisfy the condition in Formula (9). Formula (9) uses the ground truth to calculate error (similarly to SGAN-20 [28]), which is not available for unimodal prediction purposes.

## 4 Results and discussion

In this Section, we evaluate our predictive and generative models on the four datasets we proposed. We expect GAN-1 to be outperformed by both LSTM and GAN-3. In fact, the former should predict an average behavior that is more accurate than a random behavior, while the latter should find the correct behavior. The evaluation of GAN-1 is made for the only purpose of having baseline results for generative models.

The models are evaluated using both the ADE/FDE and N-ADE/N-FDE metrics on both 4-point and 12-point predictions.

Related works do not take multimodality into account (i.e. they do not produce multiple predictions) and use biased metrics; for this reason, a direct quantitative comparison between those approaches and ours does not provide a meaningful interpretation. In fact, in [31], the authors introduced a path inference method for low-frequency FCD. They have implemented and compared three methods: WSPA [32], ST-matching [33], and EMM [34]. However, the proposed method is specifically designed for cases with minimum information (i.e. latitude, longitude, and timestamp).

Altinasi et al. [3] have also studied FCD, but their paper aimed at detecting critical patterns in urban roads. Traffic patterns were defined by considering the Level of Service, which is based on segment speed on urban arterial roads. Attempts on predicting trajectories data for estimating traffic conditions from a large historical FCD are performed in [35]. The goal was twofold: on one hand the estimation of congestion zones on a large road network, on the other the estimation of travel times within congestion zones by the time-varying Travel Time Indexes (TTIs). Recently, in [36], the authors proposed a trajectory restoration algorithm, based on geometry map matching algorithms. Nevertheless, even in this case these approaches doesn not consider the multimodality.

For the aforementioned reason, we compare our GAN-3 model with an LSTM model, since the latter is the type generally used by related works. The advantage of using our GAN-3 model over those works can be assessed by comparing it with our LSTM model, using both standard metrics and normalized ones.

In Section 4.1, we show the results of these experiments, while in Section 4.2, we provide an analysis of the traffic flow, with the purpose of mapping those results into the physical world.

### 4.1 Experiments with LSTM and GAN

In the following tables, each column refers to a dataset, so that in a given column (e.g. the dataset Rome), we have the values of the error metrics for the models trained and evaluated on that given dataset.

In Tables 2 and 3 we show the values of ADE/FDE and N-ADE/N-FDE, respectively. We repeated the experiments for the three models (LSTM, GAN-1, GAN-3) and for both short predictions (4 points) and long predictions (12 points).

As expected, GAN-1 always performs the worst, because neither it minimizes the error by picking an "average behavior" like LSTM does, nor it picks the correct behavior as GAN-3 usually does.

**Table 2. Results summary (ADE/FDE).** The values in bold are the lowest on each column.

| Method | Milan | Rome | Naples | Turin |
|---|---|---|---|---|
| LSTM (4 points) | **0.0046 / 0.0071** | **0.0071 / 0.0112** | **0.0056 / 0.0087** | **0.0050 / 0.0076** |
| GAN-1 (4 points) | 0.0152 / 0.0219 | 0.0509 / 0.0570 | 0.0110 / 0.0144 | 0.0202 / 0.0314 |
| GAN-3 (4 points) | 0.0112 / 0.0164 | 0.0203 / 0.0263 | 0.0075 / 0.0090 | 0.0154 / 0.0231 |
| LSTM (12 points) | **0.0107 / 0.0191** | **0.0191** / 0.0336 | **0.0126 / 0.0221** | **0.0105 / 0.0183** |
| GAN-1 (12 points) | 0.0327 / 0.0588 | 0.0630 / 0.1023 | 0.0420 / 0.0734 | 0.0351 / 0.0570 |
| GAN-3 (12 points) | 0.0163 / 0.0244 | 0.0244 / **0.0261** | 0.0237 / 0.0531 | 0.0170 / 0.0273 |

**Table 3. Results summary (N-ADE/N-FDE).** The values in bold are the lowest on each column.

| Method | Milan | Rome | Naples | Turin |
|---|---|---|---|---|
| LSTM (4 points) | **0.0032 / 0.0051** | **0.0048 / 0.0076** | **0.0043 / 0.0069** | **0.0034 / 0.0054** |
| GAN-1 (4 points) | 0.0126 / 0.0189 | 0.0398 / 0.0406 | 0.0084 / 0.0123 | 0.0164 / 0.0276 |
| GAN-3 (4 points) | 0.0086 / 0.0123 | 0.0158 / 0.0230 | 0.0063 / **0.0066** | 0.0147 / 0.0232 |
| LSTM (12 points) | **0.0123 / 0.0220** | **0.0208** / 0.0359 | **0.0144 / 0.0255** | **0.0125** / 0.0218 |
| GAN-1 (12 points) | 0.0424 / 0.0759 | 0.0710 / 0.1314 | 0.0530 / 0.1030 | 0.0424 / 0.0814 |
| GAN-3 (12 points) | 0.0174 / 0.0294 | 0.0277 / **0.0329** | 0.0175 / 0.0321 | 0.0167 / **0.0202** |

In most cases, LSTM outperforms GAN-3, meaning that the latter is not a replacement for the former; in fact, both methods have advantages and disadvantages in different contexts.

In the 4-point predictions, LSTM outperforms GAN-3 in all cases but one: N-FDE on the Naples dataset. The reason why LSTM is better for short predictions is that uncertainty grows over time, and multimodality is just aleatoric uncertainty. As the model tries to predict more time-steps, the number of possible behaviors grows, so the "average behavior" predicted by the LSTM becomes less accurate than the correct behavior identified by GAN-3. On the other hand, when the number of time-steps is low, the LSTM prediction is more accurate, since it minimizes the square error instead of just generating plausible trajectories as the GAN does.

In the 12-point predictions, LSTM still outperforms GAN-3 on the average errors (ADE and N-ADE). GAN-3 outperforms LSTM on the Rome dataset (FDE, N-FDE) and the Turin dataset (N-FDE). Again, the reason why is GAN-3 performs better on final errors while LSTM performs better on average errors is that the cumulative uncertainty of the entire trajectory adds up to the final point. Since this only happens on the Rome and Turin datasets, we can deduce that these cities have a road graph with a highly multimodal configuration, allowing for more possible routes.

In Table 4, we show the improvement of the GAN-3 error over the GAN-1 error. The higher the ratio, the higher is the improvement. A high ratio means that GAN-3 generates better solutions than GAN-1 and it indicates multimodality, as different trajectories are generated.

We can make three main observations on this table. First, the improvement is better on long trajectories than on short ones, as expected. In fact, long trajectories have more uncertainty and there is more room for improvement with a multimodal generative approach.

Second, normalized metrics show a better improvement than standard ones. This follows directly from how we defined the GAN-3 model, which measures the distance between two trajectories using the N-ADE and not the ADE. In this way, it optimizes improvements over more high-uncertainty trajectories, as uncertainty is correlated with length and variance.

**Table 4. GAN-1 / GAN-3 ratio: The higher the ratio, the higher is the improvement of GAN-3 over GAN-1, meaning that better solutions are found by the former (a sign of multimodality).**

| Method | Milan | Rome | Naples | Turin |
|---|---|---|---|---|
| ADE (4 points) | 1.3571 | 2.5074 | 1.4667 | 1.3117 |
| FDE (4 points) | 1.3354 | 2.1673 | 1.6000 | 1.3593 |
| N-ADE (4 points) | 2.0061 | 2.5820 | 1.7722 | 2.0647 |
| N-FDE (4 points) | 2.4098 | 3.9195 | 1.3823 | 2.0879 |
| ADE (12 points) | 1.4651 | 2.5190 | 1.3333 | 1.1156 |
| FDE (12 points) | 1.5366 | 1.7652 | 1.8636 | 1.1897 |
| N-ADE (12 points) | 2.4368 | 2.5632 | 3.0286 | 2.5389 |
| N-FDE (12 points) | 2.5816 | 3.9939 | 3.2087 | 4.0297 |

Third, the best improvements are achieved in the Rome and Turin datasets. Those are the same datasets where GAN-3 outperforms LSTM, as we saw in Tables 2 and 3.

Multimodality can also be analyzed from the error growth. In Table 5, we show the N-FDE / N-ADE ratio of these experiments. A higher ratio means that the error grows faster, so the cumulative uncertainty over several time-steps is higher.

In most cases, as expected, this ratio is lower for GAN-3 experiments. It means that, although the LSTM error is generally lower than the GAN error, it tends to grow faster. One possible interpretation of this phenomenon is that LSTM and GAN-3 having different types of error: while the GAN-3 error can be associated with a less accurate model (in general, GANs are harder to train than LSTM networks), the LSTM error is due to a divergence from the actual behavior as the uncertainty increases with the number of time-steps.

As we saw in Section 2, recent works on FCD trajectory prediction are based on LSTM models that minimize the Root Mean Square Error (RMSE). From our results, we can conclude that the LSTM model is better suited for unimodal scenarios (where choices are heavily constrained) and short-term predictions, while GAN-3 works best in scenarios with high uncertainty, especially when the number of time-steps grows larger.

In these cases, finding an "average behavior", as the LSTM does, is not a satisfying solution. In fact, the more accurate the model tries to be, the worse it performs, because the average behavior diverges from any actual behavior. Our work is the first that proposes a generative approach for FCD trajectory prediction, and we have shown that we can address the problem of multimodality by generating a set of multiple realistic predictions.

## 4.2 Traffic flow state analysis

As mentioned in the introduction, traffic flow states calculated from high-resolution trajectories are an important and critical information for traffic control. Predicted trajectories

**Table 5. N-FDE / N-ADE ratio: The higher the ratio, the faster the error grows (a sign of multimodality).** The values in bold are the lowest on each column, the ones where the error is the least affected by multimodality.

| Method | Milan | Rome | Naples | Turin |
|---|---|---|---|---|
| LSTM (4 points) | 1.5719 | 1.5708 | 1.5792 | 1.5852 |
| GAN-1 (4 points) | 1.4987 | **1.0205** | 1.4668 | 1.6831 |
| GAN-3 (4 points) | **1.4420** | 1.4539 | **1.0538** | **1.5823** |
| LSTM (12 points) | 1.7918 | 1.7269 | **1.7661** | 1.7440 |
| GAN-1 (12 points) | 1.7909 | 1.8492 | 1.9424 | 1.9191 |
| GAN-3 (12 points) | **1.6864** | **1.1870** | 1.8380 | **1.2078** |

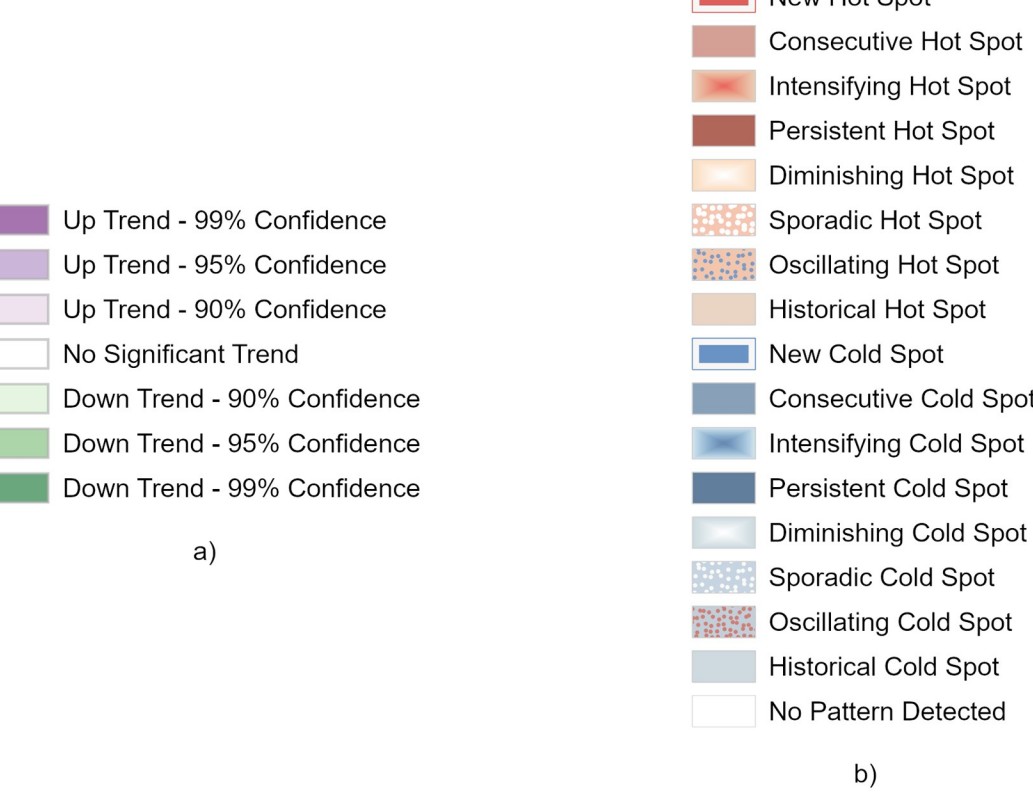

**Fig 4. Legends for trend on number of positions (a) and for hot spot analysis (b).**

generated with the methodology described in section 3 have been analyzed through the creation and analysis of space-time data cubes, exploiting the ESRI ArcGIS Pro Space Time Pattern Mining toolbox: trajectory points have been therefore inserted into a netCDF data structure, by aggregating positions into space-time bins. For all bin locations, the trend has been evaluated by counting those positions.

Two-dimensional representations of the space-time data cube attributes are a powerful tool to detect trends in space and time: in this particular framework, space-time data cube trends visualization has been exploited to compare trends in cell occupancy, based on both real data and simulated trajectories. An additional hot spot analysis, finalized to the identification of statistically significant spatial clusters of high values (hot spots) and low values (cold spots), has been performed using the Hot Spot Analysis (Getis-Ord Gi*) ESRI ArcGIS Pro tool. Fig 4 displays the legends used for displaying, while in the subsequent figures, we show the results of both the trend and hot spot analysis.

The space-time analysis has been performed on the 12-point predictions: the longer prediction is considered better fitting the needs of a traffic control center in managing traffic flows in urban contexts. The analyzed trajectories are those predicted using the LSTM network because this model produces good results on all datasets, while GAN-3 outperforms LSTM only under particular conditions, so they can be better analyzed in this context. Fig 5 displays the real positions plotted on top of the predicted ones, while Table 6 displays the distribution of planimetric distance values between real positions and the nearest predicted ones. This is a different

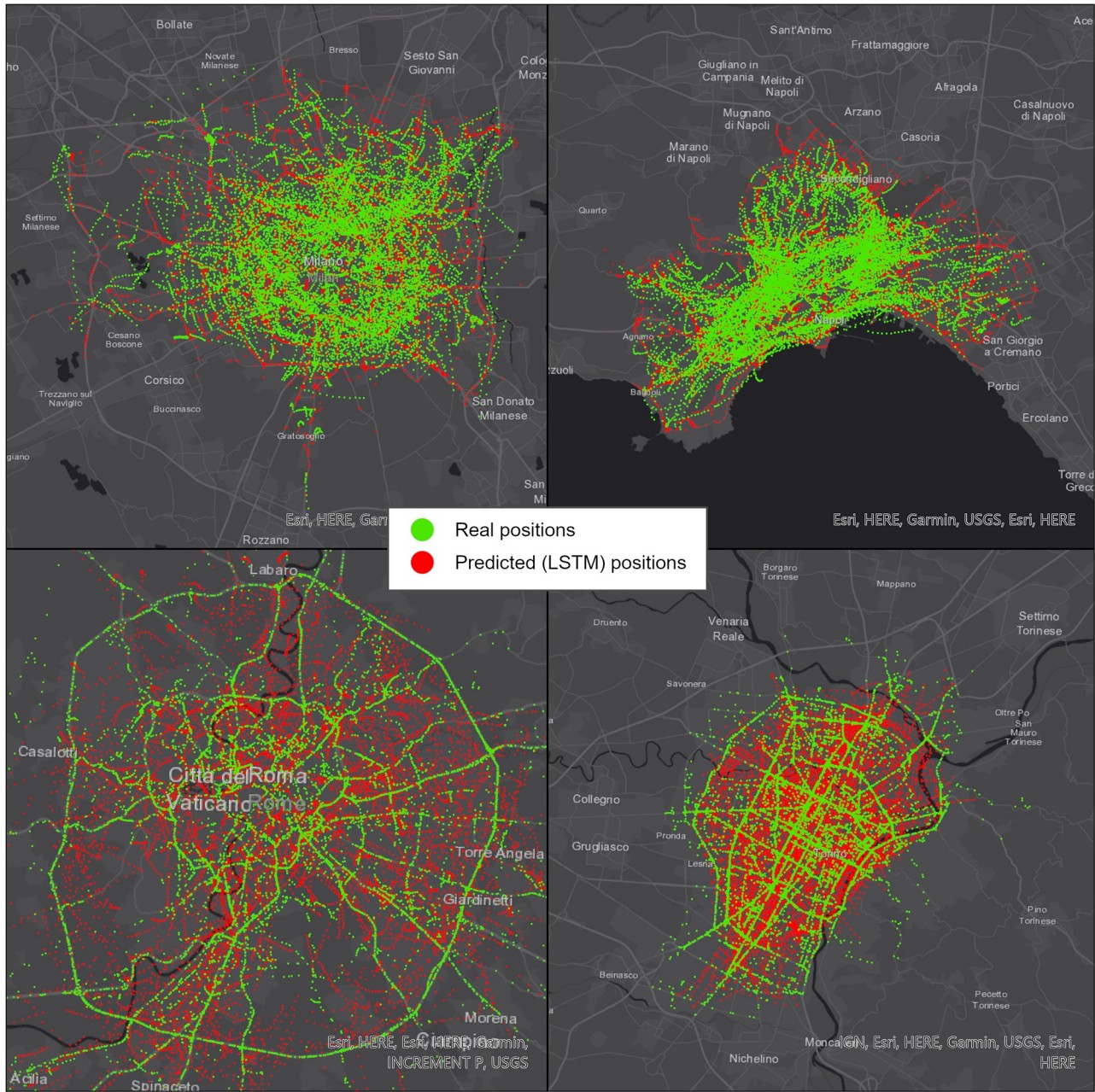

**Fig 5. Visual comparison between real and predicted (LSTM) positions in Milan (top left), Naples (top right), Rome (bottom left) and Turin (bottom right).** Republished from ESRI and HERE under a CC BY license, with permission from ESRI, original copyright 2021.

**Table 6. Position distance statistics (m) measured in the 4 cities.**

| Statistic | Milan | Rome | Naples | Turin |
|:---:|:---:|:---:|:---:|:---:|
| Mean | 60.3 | 154.4 | 84.6 | 55.5 |
| Median | 39.7 | 89.1 | 60.7 | 32.0 |
| Std. Dev. | 80.2 | 314.8 | 80.1 | 122.4 |
| Max | 1,623.8 | 7,338.2 | 856.1 | 1,958.9 |

kind of distance from the ones used to validate the predictions in Section 4.1, because this is a spatial analysis while the previous one was temporal. Unlike the previously proposed metrics, we don't use this type of distance to evaluate the models because there is no direct correspondence between real and predicted values. Nevertheless, those value distributions allow us to perform some considerations:

- There is a clear correlation between distance values and the dimension of the considered city: the size of the Rome municipality is 1,287 $km^2$, compared to 182 $km^2$ of the Milan municipality, 130 $km^2$ of the Turin municipality, and 119 $km^2$ of the Naples municipality. On a wider area, predicted positions are potentially more dispersed and this pattern is detectable in the statistics of the four cities.

- Despite being only the third in terms of geographical extension, Turin has a standard deviation value much higher than Milan (almost 30% more extended) and Naples (only 10% less extended). This higher dispersion of the values can be correlated to the particular morphology of the municipality area: the E-SE part of the municipality lies in a hilly area, with a limited number of roads, very sparse patterns, and a low number of trajectories.

Figs 6 to 9 display the results of the trend (top) and hot spot (bottom) analysis on the four cities: Milan, Naples, Rome, and Turin. Point aggregation has been made on hexagons with an height of 500 m. In all figures, the left map is obtained from the real values, while the right one is generated from the 12-point LSTM predictions.

As far as the trend analysis is concerned (top of Figs 6 to 9), overall patterns show a low degree of similarity, meaning that V2I analysis is needed in future works. Furthermore, our predictions didn't take into account information from the road graph. We can observe that predicted trajectories are not confined to existing infrastructure elements (i.e. road edges) and, therefore, are subject to be placed on areas not dedicated to vehicles, consequently creating more disperse point clouds.

This aspect is highlighted both in Fig 5 and in Table 7: in three out of the four considered cities (Milan, Naples, and Rome), the total number of non-empty cells is lower for the analysis based on real values rather than predicted ones. In the case of Rome, the difference is significant and this can be explained by the size of the city (and, consequently, of the dataset), almost one order of magnitude larger than the other three. Turin is the only city with a significant decrease in the total number of non-empty cells in the predicted scenario and this again is partially due to its morphology.

The hot spot analysis (bottom of Figs 6 to 9) shows the highest levels of correlation between real and predicted positions, even if this one too is affected by the non-confinement of the latter, leading generally to larger proportions of areas with some detected patterns.

This type of analysis is clearly conditioned by the size of the space-time cube cells: e.g., increasing hexagons resolution (i.e. decreasing the hexagon height) to 100 m, the trend comparison shows narrower differences between real and predicted values in almost all trend classes (Table 8). This is mainly explained by the fact that, reducing the size of the cells, the probability of having a relevant number of positions in each cell is lower, as well as the possibility of detecting trends: comparing Tables 7 and 8, the percentage of cells with no significant trend is in the range of 52.26–83.92% for the 500 m aggregation and 89.31–98.00% for the 100 m aggregation. Consequently, percentages of cells with some trend detected are extremely marginal when using 100 m aggregation and differences are narrower. Table 9 compares the overall accuracy over the 4 cities for both the 500 m and 100 m aggregation levels. This confirms that the smaller aggregation size provides significantly higher correspondences in the trend analysis.

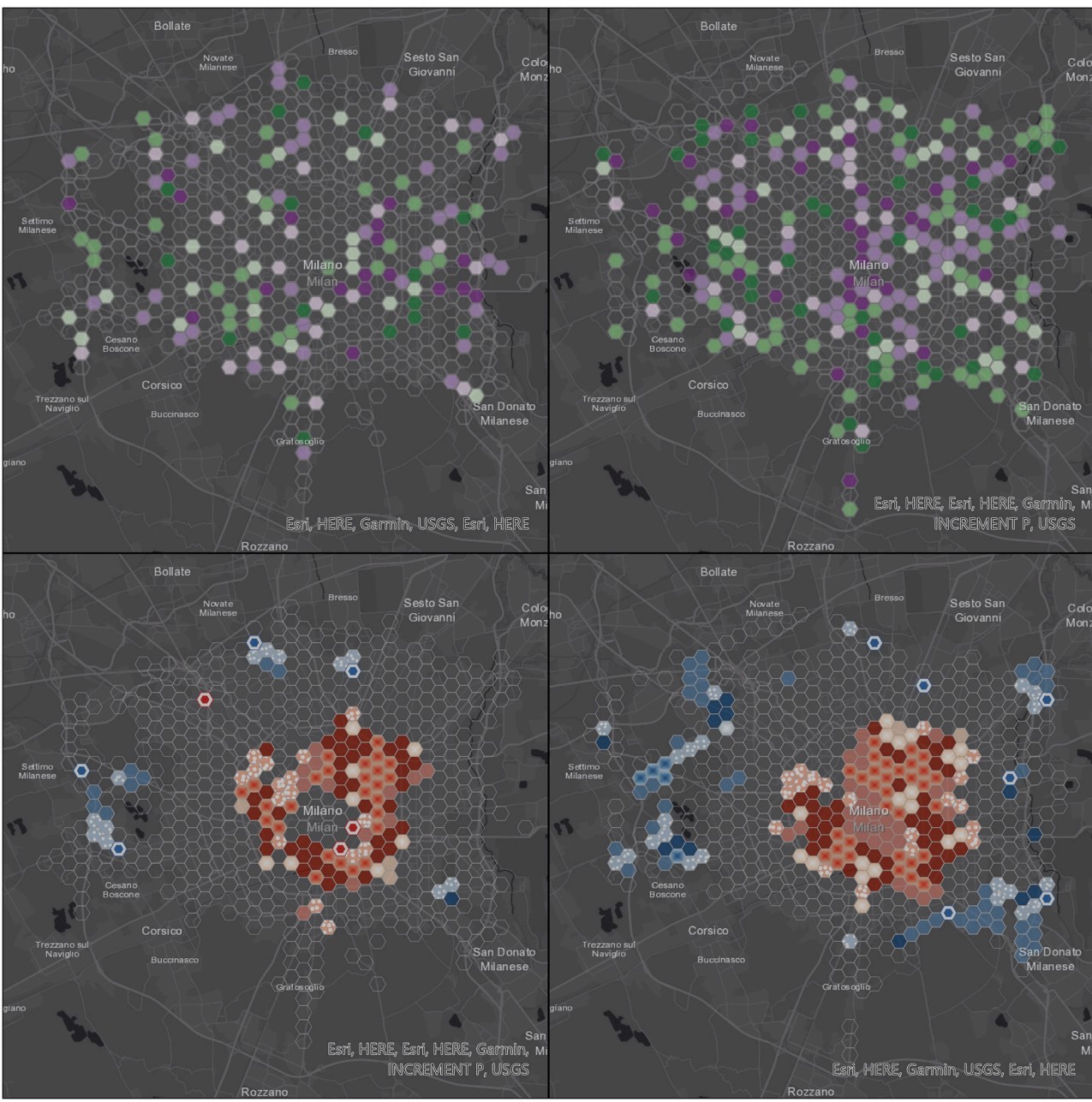

**Fig 6. Visualization of space-time data cube with 500 m resolution using the 12 points prediction over Milan area.** Republished from ESRI and HERE under a CC BY license, with permission from ESRI, original copyright 2021.

## 5 Conclusion and future works

In this work, we addressed challenges and aspects of the FCD trajectory prediction problem that have been neglected by other works in this field. We proposed new datasets with a large number of data points and multimodal scenarios, as well as new aggregate metrics that reduce bias by normalizing ADE and FDE to critical properties of the trajectories.

We addressed the problem of multimodality in a way that is missing in the current state of the art, proposing the generative model GAN-3. Although the standard LSTM model is

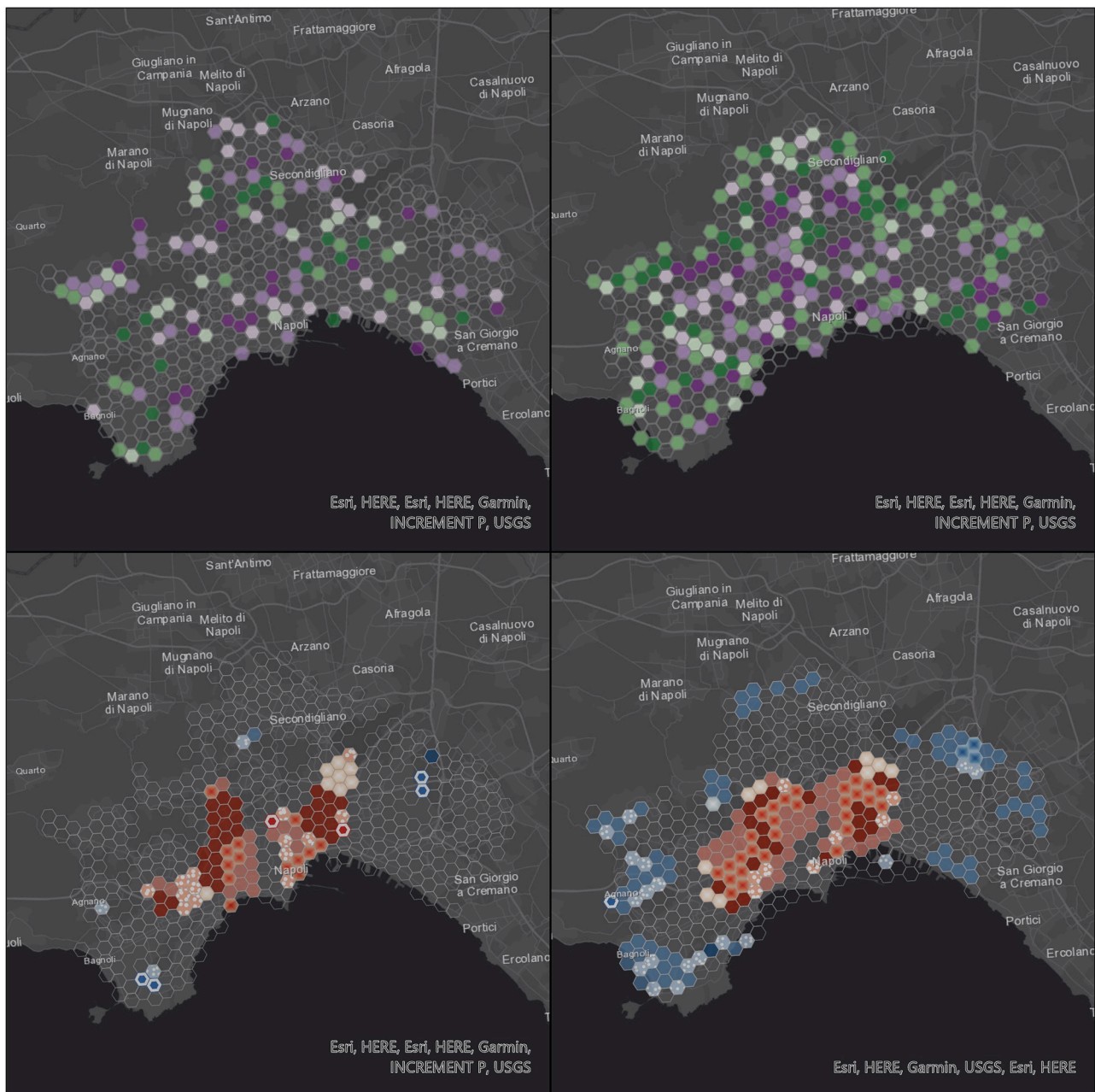

**Fig 7. Visualization of space-time data cube with 500 m resolution using the 12 points prediction over Naples area.** Republished from ESRI and HERE under a CC BY license, with permission from ESRI, original copyright 2021.

superior in unimodal scenarios, most real-world ones are multimodal and require a different approach. Our GAN-3 generates different plausible solutions instead of a single average one, and it outperforms LSTM in those cases where uncertainty is high. GAN-3 doesn't replace LSTM, as the latter is still superior in unimodal scenarios, but each method is best suited for different contexts.

Since this is the first work to use generative methods in FCD trajectory prediction, there are many possible directions for future research in the field.

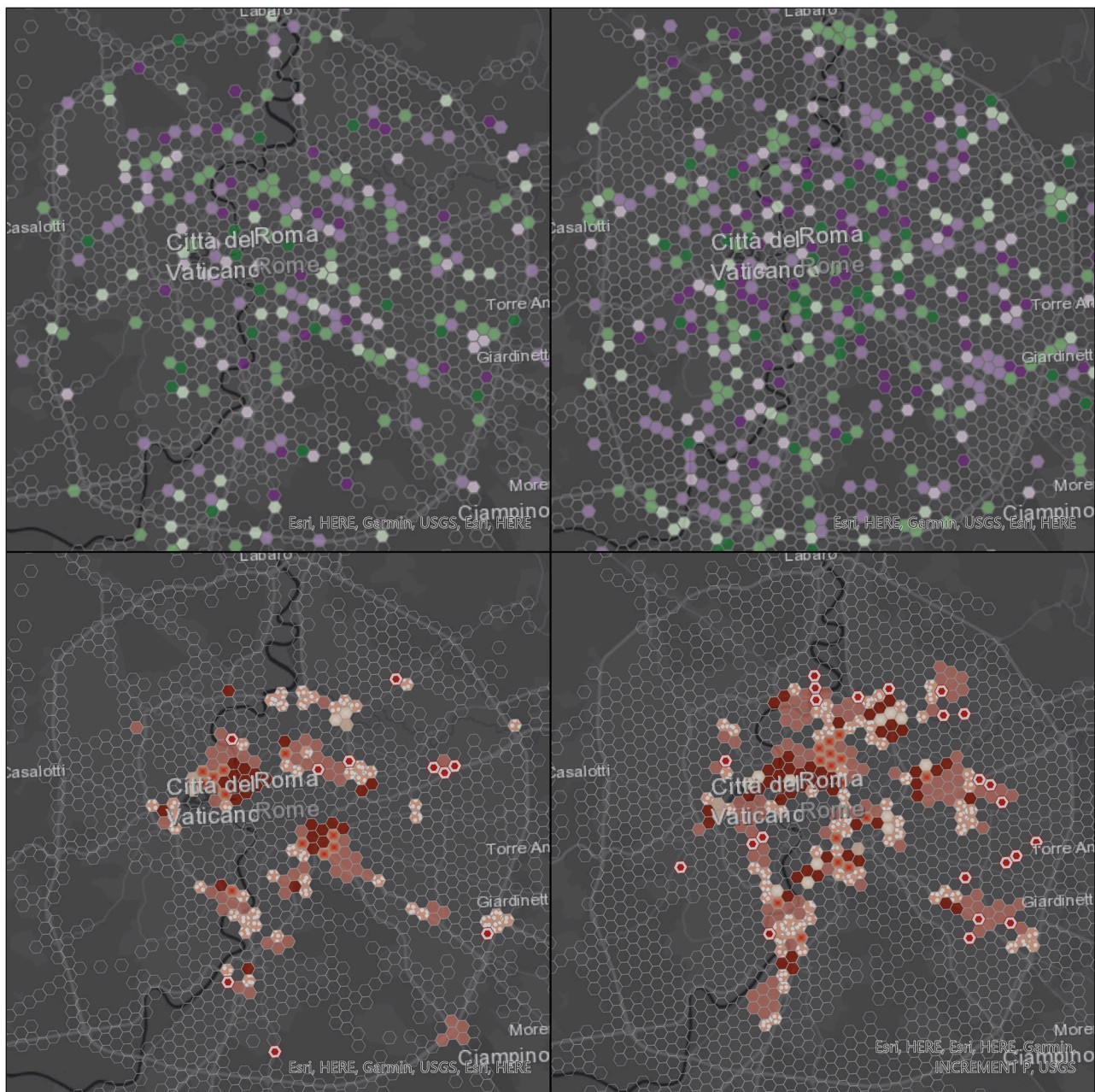

**Fig 8. Visualization of space-time data cube with 500 m resolution using the 12 points prediction over Rome area.** Republished from ESRI and HERE under a CC BY license, with permission from ESRI, original copyright 2021.

One possible improvement is the analysis of longer trajectories. This would be impossible in multimodal scenarios with a standard LSTM, but generative models should be able to tackle this challenge, enumerating different possible behaviors in a long time frame.

Another possible improvement is the study of V2V interaction that we mentioned among the challenges of FCD trajectory prediction. Different vehicles influence each other's trajectories in a major way, taking this aspect into account would largely reduce uncertainty (and the problems that arise from multimodality) and improve accuracy.

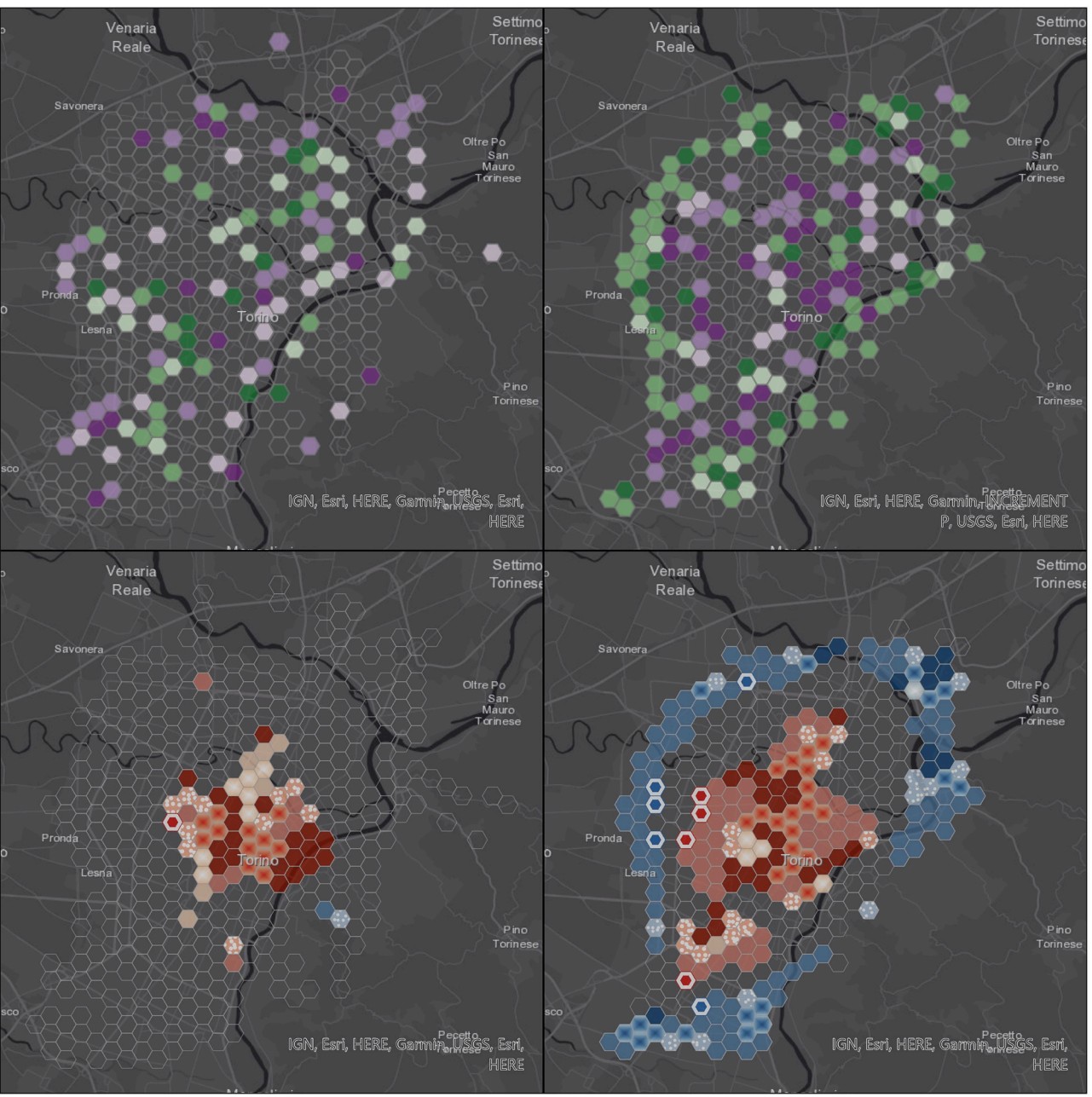

**Fig 9. Visualization of space-time data cube with 500 m resolution using the 12 points prediction over Turin area.** Republished from ESRI and HERE under a CC BY license, with permission from ESRI, original copyright 2021.

Another class of improvements would be the study of V2I interaction by making use of additional information to improve accuracy, such as the road network and related traffic restrictions, that heavily constrains the possible path a vehicle can follow. This type of information has been ignored in this work, mainly for the purpose of comparing LSTM and GAN-3 in a purely data-driven fashion. Nevertheless, a hybrid approach can bring several advantages in practical scenarios.

Furthermore, we only evaluated the models on the datasets they have been trained on, future works could include experiments that train and evaluate models on different datasets. If

**Table 7. Variation in % of the cells in each trend class in the 4 cities computed using the 500 m aggregation: Positive values are classes where real values have more occurrences than predicted ones.** In brackets the real and the predicted percentage values: for the total number of cells with at least one value, the real and the predicted total numbers are reported in brackets.

| Trend | Milan | Rome | Naples | Turin |
|---|---|---|---|---|
| Up Trend—99% Confidence | -2.71 (2.94–5.65) | -4.06 (3.90–7.96) | -1.09 (1.61–2.70) | -5.03 (2.95–7.98) |
| Up Trend—95% Confidence | -3.01 (5.86–8.87) | -0.60 (7.58–8.18) | -1.62 (4.51–6.13) | -0.77 (6.14–6.91) |
| Up Trend—90% Confidence | -0.61 (3.58–4.19) | 0.47 (5.63–5.16) | -0.53 (2.39–2.92) | 2.75 (5.68–2.93) |
| No significant trend | 14.94 (76.72–61.76) | 19.39 (71.65–52.26) | 5.48 (83.92–78.44) | 17.67 (74.32–56.65) |
| Down Trend–90% Confidence | -0.45 (3.75–4.20) | -0.19 (4.11–4–30) | 0.14 (2.62–2.48) | -0.93 (3.86–4.79) |
| Down Trend—95% Confidence | -4.47 (4.72- 9.19) | -10.94 (3.68–14.62) | -1.64 (3.67–5.31) | -9.78 (4.32–14.10) |
| Down Trend—99% Confidence | -3.69 (2.44–6.13) | -4.06 (3.46–7.52) | -0.75 (1.28–2.03) | -3.92 (2.73–6.65) |
| Total number of cells with at least one value | -0.98 (614–620) | -0.65 (462–465) | -23.59 (1,797–2,221) | 14.55 (440–376) |

**Table 8. Variation in % of the cells in each trend class in the 4 cities computed using the 100 m aggregation: Positive values are classes where real values have more occurrences than predicted ones.** In brackets the real and the predicted percentage values: for the total number of cells with at least one value, the real and the predicted total numbers are reported in brackets.

| Trend | Milan | Rome | Naples | Turin |
|---|---|---|---|---|
| Up Trend—99% Confidence | 0.22 (0.66–0.44) | 0.31 (0.59–0.28) | 0.30 (0.38–0.08) | 0.69 (0.75–0.06) |
| Up Trend—95% Confidence | 0.60 (2.46–1.86) | -0.81 (3.36–2.55) | 0.92 (1.74–0.82) | 1.62 (3.14–1.52) |
| Up Trend—90% Confidence | 0.31 (1.53–1.22) | -0.29 (1.68–1.97) | 0.38 (0.70–0.32) | 0.15 (1.93–1.78) |
| No significant trend | -2.79 (90.77–93.56) | -1.17 (90.15–91.32) | -3.74 (94.26–98.00) | -4.55 (89.31–93.86) |
| Down Trend—90% Confidence | 0.60 (1.75–1.15) | -0.14 (1.59–1.73) | 0.73 (1.06–0.33) | 1.05 (2.25–1.20) |
| Down Trend—95% Confidence | 0.86(2.39–1.53) | 0.04 (2.09–2–05) | 1.10 (1.53–0.43) | 0.63 (2.13–1.50) |
| Down Trend—99% Confidence | 0.20 (0.44–0.24) | 0.44 (0.53–0.09) | 0.31 (0.33–0.02) | 0.40 (0.49–0.09) |
| Total number of cells with at least one value | -51.63 (4,108–6,229) | -58.33 (3,391–5,369) | -49.20 (6,599–9,846) | -53.89 (3,472–5,343) |

**Table 9. Overall accuracy (%) measured on the count trends in the 4 cities.**

| Method | Milan | Rome | Naples | Turin |
|---|---|---|---|---|
| 500 m | 53.4 | 44.8 | 69.9 | 34.2 |
| 100 m | 86.7 | 84.6 | 93.8 | 75.9 |

different datasets have very different results, it means that predictions depend more on the environment than on general properties of vehicle trajectories, and that would require further research in V2I interaction.

The penetration rate of CAVS should be adequately kept into consideration because it can highly influence the performances of UTCs based on those data: Ajmar et al. in [13] provided some insights comparing floating car data (FCD) with low penetration rates with data acquired by fixed sensors, but more research studies on the impact of low penetration rates are still required. Nevertheless, CAVs penetration rates and smart road development is expected to grow systematically in the next years and the methodology proposed here can be easily scaled and implicitly benefit in accuracy by the availability of more complete data.

## Supporting information

**S1 File.**
(ZIP)

## Acknowledgments

The authors would like to thank VEM Solutions Spa a VIASAT GROUP S.p.A. company (www.vemsolutions.it) for providing FCD data used in this work.

## Author Contributions

**Conceptualization:** Luca Rossi.

**Data curation:** Luca Rossi, Andrea Ajmar.

**Formal analysis:** Luca Rossi, Andrea Ajmar, Roberto Pierdicca.

**Methodology:** Luca Rossi, Marina Paolanti.

**Supervision:** Roberto Pierdicca.

**Validation:** Luca Rossi.

**Visualization:** Andrea Ajmar.

**Writing – original draft:** Luca Rossi, Andrea Ajmar, Marina Paolanti, Roberto Pierdicca.

**Writing – review & editing:** Marina Paolanti, Roberto Pierdicca.

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
