## [Decision Letter · Decision Letter 0]

6 May 2021

PONE-D-21-10854

Vehicle trajectory prediction and generation using LSTM models and GANs

PLOS ONE

Dear Dr. Paolanti,

Thank you for submitting your manuscript to PLOS ONE. After careful consideration, we feel that it has merit but does not fully meet PLOS ONE’s publication criteria as it currently stands. Therefore, we invite you to submit a revised version of the manuscript that addresses the points raised during the review process.

Based on the comments received from the reviewers and my own observation, I recommend minor revisions for the manuscript.

We look forward to receiving your revised manuscript.

Kind regards,

Thippa Reddy Gadekallu

Academic Editor

PLOS ONE

Journal Requirements:

3. We note that Figure(s) 5-9 in your submission contain map/satellite images which may be copyrighted. All PLOS content is published under the Creative Commons Attribution License (CC BY 4.0), which means that the manuscript, images, and Supporting Information files will be freely available online, and any third party is permitted to access, download, copy, distribute, and use these materials in any way, even commercially, with proper attribution. For these reasons, we cannot publish previously copyrighted maps or satellite images created using proprietary data, such as Google software (Google Maps, Street View, and Earth). For more information, see our copyright guidelines: http://journals.plos.org/plosone/s/licenses-and-copyright.

a) You may seek permission from the original copyright holder of Figure(s) 5-9to publish the content specifically under the CC BY 4.0 license. 

4. We note that you have not provided affiliation details for author Andrea Ajmar in the manuscript title page. Please amend your list of authors in the manuscript to ensure that each author is linked to an affiliation. Authors’ affiliations should reflect the institution where the work was done (if authors moved subsequently, you can also list the new affiliation stating “current affiliation:….” as necessary).

Reviewers' comments:

Reviewer's Responses to Questions

**Comments to the Author**

1. Is the manuscript technically sound, and do the data support the conclusions?

Reviewer #1: Yes

Reviewer #2: Yes

2. Has the statistical analysis been performed appropriately and rigorously? 

Reviewer #1: Yes

Reviewer #2: Yes

3. Have the authors made all data underlying the findings in their manuscript fully available?

Reviewer #1: Yes

Reviewer #2: Yes

4. Is the manuscript presented in an intelligible fashion and written in standard English?

Reviewer #1: Yes

Reviewer #2: Yes

5. Review Comments to the Author

Reviewer #1: The paper is written good.

- What are the evaluations used for the verification of results?

- Clearly highlight the terms used in the algorithm and explain them in the text.

- Please make a comparision with existing works.

- Authors should add the most recent reference:

1) Anomaly Detection in Automated Vehicles Using Multistage Attention-Based Convolutional Neural Network, IEEE Transactions on Intelligent Transportation Systems

2) CANintelliIDS: Detecting In-Vehicle Intrusion Attacks on a Controller Area Network using CNN and Attention-based GRU, IEEE Transactions on Network Science and Engineering

Reviewer #2: • Add the advantages of the proposed system in one quoted line for justifying the proposed approach in the Introduction section.

• The motivation for the present research would be clearer, by providing a more direct link between the importance of choosing your own method.

• In the introduction, the findings of the present research work should be compared with the recent work of the same field towards claiming the contribution made. , kindly provide several references to substantiate the claim made in the abstract (that is, provide references to other groups who do or have done research in this area).

The authors can cite the following references

A multidirectional LSTM model for predicting the stability of a smart grid

Comparative analysis of machine learning algorithms for prediction of smart grid stability†

Genetically Optimized Prediction of Remaining Useful Life

Predictive model for battery life in IoT networks

6. PLOS authors have the option to publish the peer review history of their article (what does this mean?). If published, this will include your full peer review and any attached files.

Reviewer #1: No

Reviewer #2: No

---

## [Author Response · Author response to Decision Letter 0]

11 Jun 2021

Dear Dr. Thippa Reddy Gadekallu,

We would like to thank the Reviewers for their valuable review and very useful comments for improvement, which we believe have helped us to strengthen the article (PONE-D-21-10854) significantly. We have revised the manuscript (revised manuscript.pdf) to address the Reviewers' concerns, and all main changes are highlighted in red.

We believe that the improved version better clarifies the overall contribution and we hope that the revised version meets the high quality standards of your respectable Journal.

We thank you for considering our revised manuscript for publication and look forward to receiving your kind response.

As far as point 3 (figures 5-9 licensing) is concerned, the background image is derived from a web map service called Dark Gray Canvas provided by ESRI (https://www.arcgis.com/home/item.html?id=1970c1995b8f44749f4b9b6e81b5ba45). The terms of use summary states that a user may include screen captures or printed or plotted maps in academic publications (research journals, textbooks, etc.). See here for additional details: https://downloads2.esri.com/arcgisonline/docs/tou_summary.pdf

Affiliation details for author Andrea Ajmar has been added (point 4)

A complete point by point response letter is attached.

Best regards,

Marina Paolanti on behalf of all authors

Dipartimento di Ingegneria dell’Informazione (DII) Università Politecnica delle Marche

via Brecce Bianche 12, Ancona (Italy) m.paolanti@univpm.it

Response to Reviewer #1

Response: We thank the Reviewer for the comments and for the valuable suggestions. Our responses can be found in this response letter. We updated our manuscript adding new text in red (please refer to manuscript marked with changes.pdf).

R1.1: The paper is written good. What are the evaluations used for the verification of results?

Response: We are really grateful for your efforts in reviewing our manuscript, and we thank you for the positive feedback. Your comments have been fundamental to improve the manuscript, which has been reworked following your suggestions.

Concerning the verification of results in the GIS environment, we added a statistical analysis based on planimetric distance values between real and predicted values (see new table 6 and associated text). We also added a brief paragraph in the introduction to further emphasize our proposed evaluation metrics as an important contribution of this work.

R1.2: - Clearly highlight the terms used in the algorithm and explain them in the text.

Response: We added a few paragraphs in the Implementation Details section, explaining the LSTM

layers and how the Generator and Discriminator process the GAN input.

R1.3:- Please make a comparision with existing works.

Response: We added new references and highlighted the difference between those works and our

methodology in the Results and Discussions section:

1. Path inference from sparse floating car data for urban networks. Transportation Research

2. Weight-based shortest-path aided map-matching algorithm for low-frequency positioning

data

3. Map-matching forlow-sampling-rate GPS trajectories. Proceedings of the 17th ACM

SIGSPATIAL international conference on advances in geographic information systems

4. On-line map-matching framework for floating car data with low sampling rate in urban

road networks. IET Intelligent Transport Systems

5. Estimating congestion zones and travel time indexes based on the floating car data.

Computers, Environment and Urban Systems

6. A trajectory restoration algorithm for low-sampling-rate floating car data and complex

urban road networks. International Journal of Geographical Information Science

R1.4:- Authors should add the most recent reference:

1) Anomaly Detection in Automated Vehicles Using Multistage Attention-Based Convolutional

Neural Network, IEEE Transactions on Intelligent Transportation Systems

2) CANintelliIDS: Detecting In-Vehicle Intrusion Attacks on a Controller Area Network using CNN and Attention-based GRU, IEEE Transactions on Network Science and Engineering

Response:

Thanks for your wise suggestions, these references have been added in the introduction section to expand the potential scenario where similar approaches can be adopted. We also included two other references related to the use of FCD data and derived trajectories: one referring to a review on the impact of FCD penetration rates in different analysis environments and one related to a machine learning technique applied to FCD trajectories to model traffic crashes.

1. Examining the potential of floating car data for dynamic traffic management. IET Intelligent Transport Systems

2. Crash prediction based on traffic platoon characteristics using floating car trajectory data and the machine learning approach. Accident Analysis Prevention

Response to Reviewer #2

Response: We thank the Reviewer for the comments and for the valuable suggestions. Our responses can be found in this response letter. We updated our manuscript adding new text in red (please refer to manuscript marked with changes.pdf).

R2.1: Add the advantages of the proposed system in one quoted line for justifying the proposed approach in the Introduction section.

Response: We appreciate your effort and attention in evaluating our paper. We have corrected and improved the article considering your useful comments and suggestions.

We added a paragraph at the end of the introduction that summarizes the advantages of the proposed method. We also added a couple of sentences, and related references, highlighting how trajectory predictions are considered useful for both traffic management and autonomous vehicles.

R2.2: The motivation for the present research would be clearer, by providing a more direct link between the importance of choosing your own method.

Response: We added a paragraph in the introduction that summarizes the motivations already stated in the Challenges subsection.

R2.3: In the introduction, the findings of the present research work should be compared with the recent work of the same field towards claiming the contribution made. , kindly provide several references to substantiate the claim made in the abstract (that is, provide references to other groups who do or have done research in this area).

The authors can cite the following references

A multidirectional LSTM model for predicting the stability of a smart grid

Comparative analysis of machine learning algorithms for prediction of smart grid stability† Genetically Optimized Prediction of Remaining Useful Life

Predictive model for battery life in IoT networks

Response: Thank you for your suggestions. New updated references have been added to expand the literature in this domain (diverse from the one kindly suggested by you as they do not cover the specific topic of our approach) and to highlight the differences between those works and ours. Those included references are the following:

1. Path inference from sparse floating car data for urban networks. Transportation Research

2. Weight-based shortest-path aided map-matching algorithm for low-frequency positioning

data

3. Map-matching forlow-sampling-rate GPS trajectories. Proceedings of the 17th ACM

SIGSPATIAL international conference on advances in geographic information systems

4. On-line map-matching framework for floating car data with low sampling rate in urban road networks. IET Intelligent Transport Systems

5. Estimating congestion zones and travel time indexes based on the floating car data. Computers, Environment and Urban Systems

6. A trajectory restoration algorithm for low-sampling-rate floating car data and complex urban road networks. International Journal of Geographical Information Science

7. Examining the potential of floating car data for dynamic traffic management. IET Intelligent Transport Systems

8. Crash prediction based on traffic platoon characteristics using floating car trajectory data and the machine learning approach. Accident Analysis Prevention

9. Anomaly Detection in Automated Vehicles Using Multistage Attention-Based Convolutional Neural Network. IEEE Transactions on Intelligent Transportation Systems

10. CANintelliIDS: Detecting In-Vehicle Intrusion Attacks on a Controller Area Network using CNN and Attention-based GRU. IEEE Transactions on Network Science and Engineering

---

## [Decision Letter · Decision Letter 1]

15 Jun 2021

Vehicle trajectory prediction and generation using LSTM models and GANs

PONE-D-21-10854R1

Dear Dr. Paolanti,

We’re pleased to inform you that your manuscript has been judged scientifically suitable for publication and will be formally accepted for publication once it meets all outstanding technical requirements.

Kind regards,

Thippa Reddy Gadekallu

Academic Editor

PLOS ONE

Additional Editor Comments (optional):

Reviewers' comments:

Reviewer's Responses to Questions

**Comments to the Author**

1. If the authors have adequately addressed your comments raised in a previous round of review and you feel that this manuscript is now acceptable for publication, you may indicate that here to bypass the “Comments to the Author” section, enter your conflict of interest statement in the “Confidential to Editor” section, and submit your "Accept" recommendation.

Reviewer #1: All comments have been addressed

Reviewer #2: All comments have been addressed

2. Is the manuscript technically sound, and do the data support the conclusions?

Reviewer #1: Partly

Reviewer #2: Yes

3. Has the statistical analysis been performed appropriately and rigorously? 

Reviewer #1: Yes

Reviewer #2: Yes

4. Have the authors made all data underlying the findings in their manuscript fully available?

Reviewer #1: Yes

Reviewer #2: Yes

5. Is the manuscript presented in an intelligible fashion and written in standard English?

Reviewer #1: Yes

Reviewer #2: Yes

6. Review Comments to the Author

Reviewer #1: The authors have addressed my comments partly. AUthors should revisit the R1 comments and address them fully.

Reviewer #2: The authors have addressed all of my comments. The paper can can be accepted in the current format. Thank you

7. PLOS authors have the option to publish the peer review history of their article (what does this mean?). If published, this will include your full peer review and any attached files.

Reviewer #1: No

Reviewer #2: No

---

## [Editor Report · Acceptance letter]

21 Jun 2021

PONE-D-21-10854R1 

Vehicle trajectory prediction and generation using LSTM models and GANs 

Dear Dr. Paolanti:

I'm pleased to inform you that your manuscript has been deemed suitable for publication in PLOS ONE. Congratulations! Your manuscript is now with our production department. 

Kind regards, 

on behalf of

Dr. Thippa Reddy Gadekallu 

Academic Editor

PLOS ONE